# The Relation between Vitamin D Level and Lung Clearance Index in Cystic Fibrosis—A Pilot Study

**DOI:** 10.3390/children9030329

**Published:** 2022-03-01

**Authors:** Mihaela Dediu, Ioana Mihaiela Ciuca, Liviu Laurentiu Pop, Daniela Iacob

**Affiliations:** 1Paediatric Department, “Victor Babes” University of Medicine and Pharmacy Timisoara, Eftimie Murgu Square No. 2, 300041 Timisoara, Romania; dediu.mihaela@umft.ro (M.D.); pop.liviu@umft.ro (L.L.P.); 2Department of Neonatology, “Victor Babes” University of Medicine and Pharmacy Timisoara, Eftimie Murgu Square No. 2, 300041 Timisoara, Romania; iacob.daniela@umft.ro

**Keywords:** vitamin D, lung clearance index, lung function, cystic fibrosis, pediatric

## Abstract

Background: Life expectancy has increased in cystic fibrosis (CF) patients; however, the rate of mortality is still high, and in a majority of cases, the cause of death is due to respiratory deterioration. Vitamin D plays an important role in immunity and infection prophylaxis, as its deficiency is associated with frequent infections. In CF patients, a deficit of liposoluble vitamins is common, despite daily supplementation. The aim of this study is to evaluate the relation between vitamin D status and lung function expressed by lung clearance index (LCI) in patients with CF. We also assessed the relation of factors such as nutritional status, genotype, and associated comorbidities such as Pseudomonas infection, cystic fibrosis-related diabetes (CFRD), and cystic fibrosis liver disease (CFLD) with vitamin D and LCI. Methods: A cross-sectional study was conducted at the National Cystic Fibrosis Center by analyzing patients with CF who presented in our center between November 2017 and November 2019. We enrolled in the study patients diagnosed with CF, who were followed up in our CF center and who were able to perform lung function tests. Patients in exacerbation were excluded. Results: A strong negative correlation was found between vitamin D and LCI (r = −0.69, *p* = 0.000). A lower vitamin D storage was found in patients with CFLD and CFRD. Higher LCI values were found among patients with chronic Pseudomonas infection, with BMI under the 25th percentile, or with associated CFLD. Conclusion: In CF patients, vitamin D plays an important role, and its deficit correlates with an impaired LCI. Vitamin D deficit is a risk factor in patients with associated comorbidities such as CFLD and CFRD. Chronic infection with Pseudomonas, the presence of impaired nutritional status, and CFLD are associated with a prolonged LCI.

## 1. Introduction

Cystic fibrosis is a polymorphic disease with an outcome strongly driven by the associated lung disease. Even with newly discovered potentiators [1], the lung and its function must be maintained in the best condition for the greatest life expectancy and quality of life [2].

Pulmonary disease in CF has its onset in infancy [3] and may be very heterogeneous in the lung of the same individual. The early detection of pulmonary changes is very important in order to initiate the appropriate treatment [4]. Studies have shown that, even when spirometry was normal, in 30% of patients, high-resolution computed tomography (HRCT) had detected structural damage [5,6]. The lung clearance index (LCI) is a more useful parameter to evaluate the degree of lung inhomogeneity. It requires only tidal breathing and is easier to perform even for younger patients. It is an important non-invasive parameter for the early detection of lung impairment. The LCI has proven its specificity and sensibility in CF in several studies; it is considered the most reliable tool for the evaluation of respiratory status among CF children [7,8].

The inflammatory process is very intense in CF patients and has an early onset in the pediatric population. High levels of inflammatory markers are correlated with worse clinical disease. Therefore, anti-inflammatory factors might have a beneficial influence [9]. Vitamin D exposure results in the transition of a pro-inflammatory immune system to a more tolerant immune system based on the ability of calcitriol to inhibit the proliferation, differentiation, and modulation of T cells’ production of cytokines [10]. Despite daily supplementation, patients with cystic fibrosis have low levels of vitamin D [11].

Besides well-known factors influencing the CF patient’s lung function, such as infections, genotype, or nutrition, a more recently study proved the impact of vitamin D through its effects on infection and bone metabolism and immune modulator roles [12].

Initially described as a “vitamin”, calcitriol is recognized today as a hormone synthetized by the human body, which acts on its target organs through some nuclear receptors (VDR). Studies from the past two decades have shown that vitamin D is not involved only in phospho-calcium metabolism and bone mineralization, but that it also plays a role in cardiovascular disease [13], depression [14], obesity [15], and infectious disease [16].

Vitamin D also plays an important role in the innate immune system and adaptive immune system. Despite intensifying the antimicrobial activity of macrophages and monocytes, the calcitriol—VDR—X retinoid receptor complex directly activates the transcription of some antimicrobial peptides such as β2-defensine (DRFB) and cathelicidin antimicrobial peptide (hCAP18) [17,18,19]. 1-α-hydroxylase and VDR are also expressed at T and B cells’ surfaces [20] or in the lung epithelium [21]. Calcitriol has a direct effect on B cells homeostasis, including on memory inhibition and plasmatic cells production, and on B lymphocytes apoptosis [12,22,23].

Two recent articles, one meta-analysis [24], and a systematic review [25] pointed out the importance of vitamin D supplementation in the prevention of infection with SARS-CoV2 or reducing the risk for severe forms. The mechanisms involved were the role of vitamin D as a supportive supplement for the immune system and the anti-inflammatory role of vitamin D, thus preventing the cytokine storm responsible for ARDS syndrome.

The aim of this paper is to evaluate the relation between vitamin D status and lung function, assessed by the lung clearance index in patients with cystic fibrosis. We also assessed the relation of several factors such as nutritional status, genotype, and associated comorbidities such as chronic Pseudomonas infection, CFLD, and CFRD, with vitamin D and LCI.

## 2. Materials and Methods

### 2.1. Design and Setting

A cross sectional study was conducted at the National Center of Cystic Fibrosis, analyzing CF patients that attended the Center during the period November 2017–November 2019.

### 2.2. Inclusion and Exclusion Criteria

The inclusion criteria were: patients diagnosed with typical cystic fibrosis according to consensus guidelines from The Cystic Fibrosis Foundation [26]; patients agreeing to participate in the study; patients fit to perform the tidal breathing that is necessary for multiple breath wash out (MBW) maneuvers; and forced expiration for spirometry. Patients and their parents signed informed consents to be enrolled in the study. CF patients with clinical exacerbation or who were not able to perform the respiratory test up to the end, together with patients with other associated malabsorption causes such as celiac disease or inflammatory bowel disease, were excluded.

### 2.3. Data Collected

Besides anthropometric measures such as weight, height, and BMI, pulmonary function and biological investigation (sputum and blood sample) were analyzed for each patient. Data regarding age, sex, genotypes, and associated comorbidities (chronic infection with Pseudomonas, CFLD, CFRD) were extracted from the center’s records data base. Each patient performed CT; CT scores were calculated with a modified Bhalla cystic fibrosis score for HRCT [27]. The study was approved by the Ethics Committee of Clinical County Hospital Timisoara, Romania (131/01.11.2017). The approval date was 1st November 2017.

Lung function was evaluated using the LCI as an endpoint (performed through MBW). A QUARK PFT real-time gas-analyzer machine (Cosmed, It) was used to determinate the LCI using a compressed gas, nitrous oxide system (79% N_2_, 5%CO_2_, 16%O_2_, BAL.). The washout was considered complete after 7 min of rebreathing had occurred (with oxygen compressed to 99.99% O_2_) or when the final concentration of expired N_2_ was below 1.5–2.5% for three consecutive breaths. The exhaled N_2_ represented the patient’s initial lung volume, which allowed the measurement of FRC to be calculated.

Guided spirometry was performed according to the American Thoracic Society and ERS standards using an electronic spirometer (Carefusion MicroLab Spirometer). The spirometry endpoints parameters used for the detection of obstruction were forced expiratory volume in 1st second (FEV_1_) and forced expiratory flow 25–75% (FEF_25–75_). Children performed at least three spirometry tests, and the best values of each patient were considered for this study. The results were expressed as percentages predicted to age, height and weight, and the Global Lung Function Initiative 2012 reference equations were used to calculate predicted parameter value [28].

The serum level of 25-OH vitamin D was quantitatively measured using liquid chromatography–tandem mass spectrometry and was expressed in ng/mL. A 25-OH vitamin D level above 30 ng/mL was considered normal, a level between 20–29.9 ng/mL was considered insufficient, and a level below 20 ng/mL was considered vitamin D deficient, according to The Cystic Fibrosis Foundation [29].

Vitamin D was dosed at the end of the winter months to exclude bias from the solar exposure synthesis of vitamin D. All patients received supplementary liposoluble vitamins including vitamin D, according to the guidelines.

### 2.4. Statistical Analysis

Data were analyzed using IBM SPSS Statistics 26. Data distribution was assessed using the Shapiro–Wilk test. The results of descriptive statistics are presented as median and interquartile range, respectively, mean value, and standard deviation (SD). The Spearman correlation test was used to evaluate the correlation between the LCI and our variables. We used as a correlation scale Guildford’s (1973) rule of thumb [30] to interpretate the r—correlation coefficient: <0.2 negligible correlation, 0.2–0.4 low correlation, 0.4–0.7 moderate correlation, 0.7–0.9 high correlation, and >0.9 very high correlation.

The Mann–Whitney U test and the Kruskal–Wallis H test were used to compare the LCI values between our different groups and categories of patients. The cut-off value for the *p* statistic value was 0.05.

## 3. Results

### 3.1. Patients’ Characteristics

We screened 97 patients; 57.7% (56) met our criteria and were included in our study. Table 1 summarize all patients’ characteristics by age group. Except for FEV_1_ and FEF_25–75_, all quantitative data had a non-parametric distribution. The median (IQR) age was 11 years [7.00, 15.00], and 53.6% of the patients were boys. Additionally, 46.4% of patients were F508del homozygote, and 44.6% had chronic infection with Pseudomonas aeruginosa, 57.1% CFLD, and 19.6% CFRD.

A median 25-OH vitamin D level of 18.13 ng/mL [9.68, 28.93], which was consistent with the deficiency category, was found in our patients, as well as a prolonged LCI of 11.58 [7.88, 19.83], suggestive of impaired lung function.

### 3.2. Vitamin D Status

In our group, only 16.1% of patients had a normal vitamin D status, 33.9% of them had an insufficient level, and 50% had vitamin D deficiency.

Patients with a BMI below the 25th percentile were 3.52 times (95% CI: 0.781 to 15.948) more likely to have a vitamin D level less than 30 ng/mL than patients with a normal BMI percentile. We found a significant difference between median vitamin D level in patients with BMI above the 25th percentile and those with a BMI < 25th percentile (13.4 vs. 28.9, *p* = 0.000) (Appendix A) (Table 2).

No statistically significance differences were found between the median vitamin D level in patients over 18 years vs. below 18 years (12.21 vs. 20.26, *p* = 0.087) (Appendix A) or according to patients’ F508del genotype, homozygote vs. non-homozygote (15.5 vs. 23.7, *p* = 0.083) (Appendix A) (Table 2).

A vitamin D level < 30 ng/mL was a risk factor for impaired lung function expressed by an FEV_1_ < 80% (RR = 4.97, 95% CI: 0.77 to 31.16) and an FEF_25–75_ < 80% (RR = 1.34, 95% CI: 0.73 to 2.46), respectively.

A low level of vitamin D was an important risk factor for CFLD (RR = 2.87, 95% CI: 0.83 to 9.93) and associated CFRD (RR = 1.9, 95% CI: 0.279 to 13.166) in our group study (see Table 3).

A vitamin D level < 30 ng/mL does not represent a risk factor for chronic infection with Pseudomonas (RR = 1) (Table 3).

### 3.3. LCI Variability

A comparison of the mean of the LCI’s variable distribution, comparing patients according to their nutritional status, chronic Pseudomonas infection, impaired lung function, and genotype presence of CFLD and CFRD, was carried out.

The LCI value tends to be higher, with a statistically significant difference in patients with poor nutritional status, CFLD, chronic Pseudomonas infection, impaired lung function, or in patients with an F508del homozygote mutation (Appendix A) (Table 4).

In patients with low vitamin D levels, the LCI was higher than among patients with normal values 12.3 vs. 7.8, *p* = 0.019 (Figure 1).

Patients with a BMI < 25th percentile had a higher LCI than those patients with a BMI > 25th percentile, 14 vs. 8, *p* = 0.001 (Appendix A).

Additionally, patients with CFLD had an increased LCI of 14.05 compared to non-CFLD patients, where the median LCI was 8.67, *p* = 0.002 (Appendix A).

No statistically significant difference was found in the LCI value of patients with CFRD and those without CFRD (Appendix A).

An increased LCI median value of 18.8 was found, as expected, among patients with chronic infection with Pseudomonas compared to patients without chronic Pseudomonas infection, LCI = 9.1, *p* = 0.001 (Appendix A).

However, statistically significant discrepancies were found in the LCI value among the four categories of age groups (H = 14.184, df = 3, *p* = 0.003) (Figure 2).

### 3.4. Correlations of 25-OH Vitamin D

Evaluating the correlation between vitamin D level and the LCI value, we found a significant moderate negative correlation between the LCI and vitamin D (r = −0.69, *p* = 0.000) (Figure 3).

A negative correlation of vitamin D was found with spirometry parameters FEV_1_ (r = −0.588, *p* = 0.000) and FEF_25–75_ (r = −0.577, *p* = 0.000), respectively, and with CT Bhalla score (r = −0.605, *p* = 000).

## 4. Discussion

This is the first study that evaluates the relation between LCI and vitamin D, although previous studies have evaluated vitamin D status among CF patients [9,27,28,29]. Moreover, this is the first Romanian study evaluating lung function expressed by the LCI in cystic fibrosis patients showing that a lower vitamin D level is associated with impaired lung function, expressed by a prolonged LCI. A low vitamin D level (<30 ng/mL) was associated with a higher LCI value (p = 0.019), compared to patients with a normal vitamin D level (>30 ng/mL). This finding is supported by the significant negative correlation found between LCI and vitamin D (r = −0.69, *p* = 0.000), suggesting that an improved vitamin D level would be mandatory for better lung function, as proposed by other studies. Vitamin D level was also correlated with structural deterioration as expressed by a modified CT Bhalla score (r = −0.605, *p* = 0.00). Despite daily vitamin D supplementation, only 16.1% of patients had a normal vitamin D level, probably consecutive to malabsorption, poor adherence to supplements or their increase used [31], and a lack of physical activity, which is reported by other studies [32]. Another explanation might be the low BMI of our patients, which is associated with vitamin D deficiency. Patients with a BMI < 25th percentile are 3.52 times (95% CI: 0.781 to 15.948) more likely to have a vitamin D level deficit due to malabsorption [33]. This finding is supported by the lower vitamin D level that we found in patients with a BMI under the 25th percentile, compared to those with BMI > 25th percentile, 13.4 vs. 28.9 (*p* = 0.000).

The theory that a lower vitamin D level influences lung function in a negative way is also supported by the negative correlation found between FEV_1_ and FEF_25–75_ and vitamin D level. This is similar to findings reported by other studies [34], although this relation was questioned by others [35,36].

Vitamin D deficit was associated with an increased risk of patients developing CFLD (RR = 2.87) and CFRD (RR = 1.91); the relation could be reciprocal, as in CFLD, the vitamin D absorption could be decreased.

In our group, a vitamin D level lower than 30 ng/mL was not a risk factor for patients with a chronic infection with Pseudomonas. This finding is contrary to studies that underlined the anti-infectious role of vitamin D [16,24,25].

We found a difference between vitamin D level in patients above and under 18 years old (12.21 vs. 20.26, *p* = 0.087). Non-compliance to treatment for patients over 18 years, after they are no longer under parents’ supervision, might be an explanation for this finding. Another interpretation for the lack of statistically significant difference could be the disparity between populations transitioned to adult care, a relatively small number of up to 10 patients, compared to the pediatric population, where 46 patients were included in the study.

It appears that the F508del genotype does not affect vitamin D absorption; no statistically significant difference was found between F508del homozygote and heterozygote patients (15.5 vs. 23.7, *p* = 0.083).

LCI values were higher among older patients (H = 14.184, df = 3, *p* = 0.003), probably due to a progressive deterioration in lung function, although the LCI is a very good parameter to evaluate lung function because it is not influenced by age, as published before [37].

This study reports that, in the presence of CFLD, lung function is more deteriorated, as expressed by a higher LCI value (*p* = 0.000). A previous study [2] has proved that the presence of co-morbidities such as CFLD and CFRD is unrelated to lung function, as expressed by spirometry. Although a recent study about abnormal glucose tolerance and lung function in cystic fibrosis patients had concluded that patients with abnormal glucose tolerance or even with CFRD [37] have worse lung function [38], these findings were not supported by our study. The difference between the LCI value in patients with CFRD does not differ statistically from the LCI value in patients without CFRD (*p* = 0.140), probably due to the small number of patients with CFRD included in our study, which would be a study limitation, although for a rare disease such as CF, the patients’ number is acceptable.

Patients with a chronic Pseudomonas infection had a statistically significant higher LCI (*p* = 0.001), as expected and previously reported in other studies [39,40,41], and a significantly lower D vitamin serum level compared to non-infected patients, as in previous studies [42,43]. This finding supports the hypothesis of an antimicrobial effect of vitamin D in patients with normal vitamin D storage.

Homozygote patients for F508del mutation also had a higher LCI compared to heterozygote patients for F508del mutation or those with non-F508del mutation (*p* = 0.030) and a low vitamin D level.

In our study, a lower BMI was associated with a higher LCI (*p* = 0.001); poor nutritional status was associated with worse lung function and a decreased vitamin D level, as stated in previously published journals [2,44].

Our data confirm, consistent with previous papers, that lung function is influenced by the level of vitamin D, and this conclusion is due to the fact that lung function was evaluated by LCI, a more sensitive parameter of lung function in cystic fibrosis. Nevertheless, larger, longitudinal studies, with the inclusion of vitamin D supplementation quantity, adherence evaluation, and physical activity measurements, would be necessary for a better evaluation of the vitamin D effect on the lungs of CF patients.

## 5. Conclusions

In our study, vitamin D deficiency correlates with impaired lung function in CF patients, expressed by a prolonged LCI. Vitamin D deficit is a risk factor in patients with associated comorbidities as CFLD and CFRD. Chronic infection with Pseudomonas, the presence of impaired nutritional status, and CFLD are also associated with deteriorated lung function.

## Figures and Tables

**Figure 1 children-09-00329-f001:**
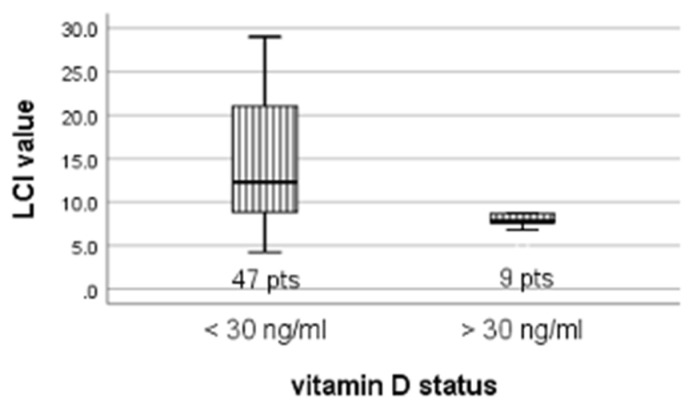
LCI variation according to vitamin D status.

**Figure 2 children-09-00329-f002:**
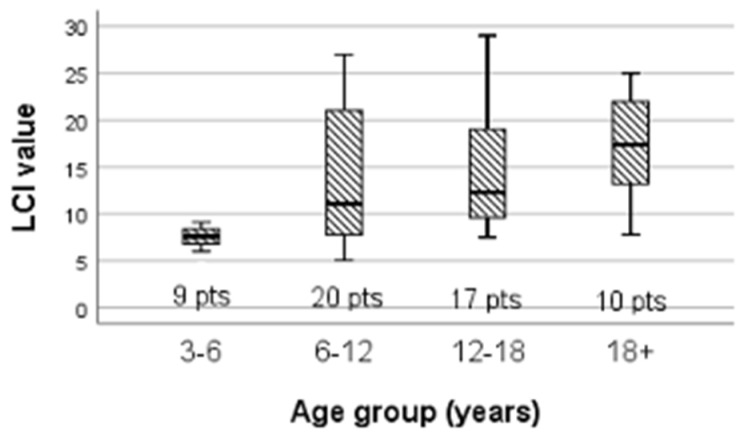
LCI variation according to age group.

**Figure 3 children-09-00329-f003:**
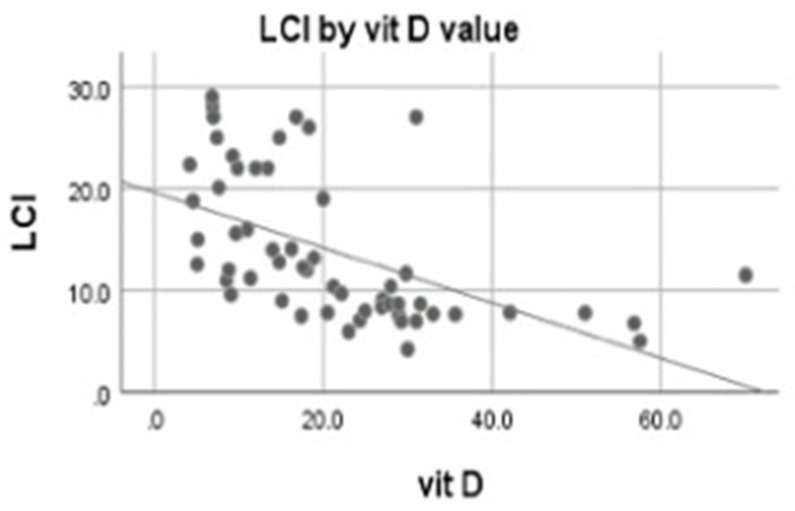
LCI variation by vitamin D level.

**Table 1 children-09-00329-t001:** Patients’ characteristics by age category.

	All Patients (N = 56)	3–6 Years (N = 9)	6–12 Years (N = 20)	12–18 Years (N = 17)	18+ Years (N = 10)
Age (years)	11 (7.00, 15.00)	5 (4.00, 5.00)	9 (7.00, 10.00)	15 (13.50, 15.00)	19 (10, 20)
BMI percentile <25th	33 (58.9%)	2 (22.2%)	9 (45%)	13 (76.5%)	9 (90%)
CFLD	32 (57.1%)	2 (22.2%)	11 (55%)	12 (70.6%)	7 (70%)
CFRD	11 (19.6 %)	0 (0%)	3 (15%)	6 (35.3%)	2 (20%)
CT score	36.5 (18.25, 57.5)	8 (7, 22)	30.5 (18.5, 50)	44 (33.5, 64)	61.5 (17, 72)
FEV_1_%	74.3 ± 21.7	86.33 ± 11.5	75.4 ± 22.4	74.35 ± 23.1	61.2 ± 20.7
FEF_25–75_%	62.84 ± 27.5	75.56 ± 15.2	63.10 ± 26.4	65.82 ± 30.4	45.8 ± 28.4
Sex					
female	26 (46.4%)	4 (44.4%)	9 (45%)	8 (47.1%)	5 (50%)
male	30 (53.6%)	5 (55.6%)	11 (55%)	9 (52.9%)	5 (50%)
Genotype F508del					
homozygous heterozygous non-F508del mutation	26 (46.4%) 26 (46.4%) 4 (7.2%)	0 (%) 8 (88.9%) 1 (11.1%)	9 (45%) 9 (45%) 2 (10%)	11 (64.7%) 5 (29.4%) 1 (5.9%)	6 (60%) 4 (40%) 0 (0%)
LCI	11.58 (7.88, 19.83)	7.6 (6.4, 8.765)	11.1 (7.78, 21.05)	12.3 (9.135, 22)	17.38 (13.2, 22.00)
Pse. chronic infection	25 (44.6%)	1 (11.1%)	5 (25%)	10 (58.8%)	9 (90%)
Vitamin D ng/mL	18.13 (9.68, 28.93)	27 (24.635, 43.39)	17.53 (11.7, 30.4)	17.4 (7.15, 24.6)	12.21 (9.63, 18.9)

SD: standard deviation; BMI: body mass index; CFLD: cystic fibrosis liver disease; CFRD: cystic fibrosis-related diabetes; CT: computed tomography; FEV: forced expiratory volume; FEF: forced expiratory flow; LCI: lung clearance index; Pse.: Pseudomonas.

**Table 2 children-09-00329-t002:** Vitamin D status (Mann–Whitney U test).

	Median Vit D	Mean Rank Vit D	Mann-Whitney U	*p* Value
BMI percentile <25th vs. >25th	13.4 vs. 28.9	22.1 vs. 37.6	169.5	0.000
Age group >18 years vs. <18 years	12.2 vs. 20.3	20.5 vs. 30.2	150	0.087
F508del mutation homozygote vs. non-homozygote	15.5 vs. 23.7	24.4 vs. 32	284.5	0.083

**Table 3 children-09-00329-t003:** Relative Risk for vitamin D < 30 ng/mL.

	RR	95% CI	
		Lower Bound	Upper Bound
CFLD	2.87	0.83	9.94
CFRD	1.92	0.28	13.17
Pse. chronic infection	1.00	0.45	2.23
FEV_1_ < 80%	4.97	0.77	31.16
FEF_25–75_ < 80%	1.34	0.73	2.46

RR: relative risk; CI: confidence interval; Pse.: Pseudomonas.

**Table 4 children-09-00329-t004:** LCI variability (Mann–Whitney U test).

	Median LCI	Mean Rank LCI	Mann-Whitney U	*p* Value
BMI percentile				
<25th vs. >25th	14 vs. 8	34.5 vs. 19.9	182.5	0.001
CFLD yes vs. no	14 vs. 8.7	34.4 vs. 20.6	194.5	0.002
CFRD yes vs. no	12.6 vs. 11	35 vs. 26.9	176.0	0.140
FEV_1_ < 80% yes vs. no	15.6 vs. 8.6	36.2 vs. 21.3	183.0	0.001
FEF_25–75_ < 80%				
yes vs. no	13.6 vs. 7.8	33.3 vs. 16.5	128.0	0.000
Genotype F508del				
homozygote vs non-del homozygote	12.6 vs. 9.4	33.6 vs. 24.1	258.0	0.030
Pse. chronic infection				
yes vs. no	18.8 vs. 9.1	36.6 vs. 21.9	184.0	0.001
Vitamin D status				
<30 ng/mL vs. >30 ng/mL	12.3 vs. 7.8	20.7 vs. 16.8	106.5	0.019

LCI: lung clearance index; BMI: body mass index; CFLD: cystic fibrosis liver disease; CFRD: cystic fibrosis-related diabetes; FEV: forced expiratory volume; FEF: forced expiratory flow; Pse.: Pseudomonas.

## Data Availability

Data available on request from the first author.

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
