# Peer review of "The Relation between Vitamin D Level and Lung Clearance Index in Cystic Fibrosis—A Pilot Study"

_children, 2022, doi:10.3390/children9030329_

Round 1
Reviewer 1 Report
The relation between vitamin D level and lung clearance index in cystic fibrosis – a pilot study
The review article by Dediu and colleagues aimed to claim the importance of vitamin D concentrations may correlated with lung clearance index and serve as a clinical indicator of cystic fibrosis. The authors compared vitamin D levels with some routinely tested factors (BMI, CFLD, CFRD, FEV, etc.). and shows the importance of vitamin D in specific populations. A trade-off effect of vitamin D concentration was also demonstrated when compared with LCI, respectively. The article also explores site-specific changes in vitamin D. The article suggests that vitamin D will correlate with pulmonary clearance index. However, this manuscript requires additional expansion and comparison. They need to recruit more experimental results and comprehensive analysis. There are some flaws in the current version, the whole manuscript needs to be reorganized. The following issues must be addressed:
Major comments:
- Although the authors attempted to claim vitamin D levels and lung clearance indices in cystic fibrosis patients, they were particularly concerned with children or early-onset populations. They did not provide evidence that children and adults have different levels of vitamin D. I am not sure if vitamin D levels can only be monitored in children, or if it applies to all ages diagnosed with cystic fibrosis.
- I agree that vitamin D plays a role in certain immune functions. They can regulate some antimicrobial effects or enhance macrophages populations. As they describe, vitamin D should form a complex with its receptor or trigger downstream signaling. They should perform or discuss these factors in this study.
- Insufficient information on vitamin D mutation events and BMI issues. It is only briefly mentioned in the discussion, and detailed dependencies need to be written carefully. The current evidence represents only a marginal association.
- In their conclusion section, this result seems to be taken for granted, can they highlight the novelty of the study or where is the most important focus? Does this mean this will become a regular inspection item or can replace the current inspection protocol?
- In their figures, they revealed some boxplots to indicate LCI between vitamin D levels, BMI, pseudomonas chronic infection, CFLD, CFRD and age. But they lack boxplots between F508del homozygotes and non-F508del homozygotes. They also lacked boxplots between homozygous, heterozygous and wild-type of vitamin D levels.
- Vitamin D is closely related to blood calcium concentration or absorption. Does it need to be explored in this study? In addition, there are some immune-related components or surface markers that need to be detected. Such evidence would make the study more complete.
Minor comments:
- Please add the case number to each group in their boxplots.
- Half of the references were more than five years old. This information needs to be updated.
Reviewer 2 Report
Comments to the Authors
Paper ”The relation between vitamin D level and lung clearance index in cystic fibrosis – a pilot study” reports results from a cross-sectional study from the National Cystic Fibrosis Centre in Romania assessing the relationship between vitamin D and lung function in patients with cystic fibrosis. The authors correlate vitamin D level with lung function parameters (from MBW and spirometry), CT scores, Pseudomonas aeruginosa infection, genotype, demographic data. The study reports that higher LCI values were found among patients with Pseudomonas chronic infection and cystic fibrosis liver disease. A lower vitamin D was a risk factor for patients with cystic fibrosis liver disease and cystic fibrosis-related diabetes. The authors conclude that in CF patients, vitamin D plays an important role and its deficit correlates with an impaired LCI.
It is well known that vitamin D plays a crucial role in CF and its deficiency is most common in patients with severe disease. It may be due to various reasons, ranging from insufficient supplementation, malabsorption to non-compliance.
There are some points needing clarification.
Major comments:
- Abstract section should be modified to be more informative.
- - Line 14: The authors describe the study aim: ”...children with cystic fibrosis”. However in Table 1 10 patients 18+ were described. Please specify the study population: children? Or children and adults?
- - Line 14-15: We aimed to evaluate the relation between vitamin D and lung function (what tests?) in children with cystic fibrosis – but in Materials and Methos as well as Results sections the authors also describe the biological investigation – not only blood but also sputum, assessing CT scores – these should be mentioned in Abstract section.
- Line 17: Were these the only inclusion and exclusion criteria? Could even infants with cystic fibrosis be included in the study?
- - Lines 20-22: A lower vitamin D level was a risk factor for patients with cystic fibrosis liver disease and cystic fibrosis related diabetes – in my opinion, is not a result but a conclusion
- Lines 66-67: The authors describe that the paper aims to evaluate the relationship between vitamin D status and lung function assessed by lung clearance index in patients with cystic fibrosis. However, the other factors were also assessed in the study: results of spirometry, sputum cultures, genotype, CFRD, CFLD….. I suggest adding this information
- Design and Setting section: lines 69-72. It would be advisable to add a brief description of CF patients who are under the care of the National Centre of Cystic Fibrosis. How many patients are looked after? is it a children's center? for adults? both?
- Lines 76-77: what about spirometry? Was the ability to perform spirometry also one of the inclusion criteria? Please specify.
- Line 78: What kind of "other malabsorption causes" made patients excluded from the study? please specify.
- Line 83: when were CTs performed? Has any time criterion been adopted? in 3-6-12 months?
- Line 84: please specify when the approval of the Ethics Committee was obtained (document number and year)
- Results section: line 121: Why as many as 41 patients did not meet the inclusion criteria? what criteria? please describe it
- Table 1 and Table2: “Pse.colonization” but in line 125 is chronic infection with Pseudomonas aeruginosa. There is a difference between colonization and chronic infection. Please specify and provide the definition based on which the bacteriological status was assessed
- There are 8 figures in the paper. I propose to put the most important data on 1-2 figures, describe the remaining data or place the figures in additional materials
- In Conclusions section (lines 253-255) I suggest modifying the sentence: “Lung function stated by impaired LCI was negatively correlated with the low levels of vitamin D in our study, expressing the requirement for regular control and an increased vitamin D supplementation among CF patients with its deficiency.”
Minor comments:
- Lines 10-11: In order to clarify the reason of death in CF I suggest adding respiratory disease or complications
- Line 31: I would suggest emphasizing that: Pulmonary disease in CF is started from the infancy period
- Lines 35-38: I propose to divide this very long sentence into 2 or 3 shorter ones
